# Immune Response and Effects of COVID-19 Vaccination in Patients with Lung Cancer—COVID Lung Vaccine Study

**DOI:** 10.3390/cancers15010137

**Published:** 2022-12-26

**Authors:** Ainhoa Hernandez, Marc Boigues, Eudald Felip, Marc Cucurull, Lucia Notario, Anna Pous, Pere Torres, Marta Benitez, Marina Rodriguez, Bibiana Quirant, Margarita Romeo, Daniel Fuster, Teresa Moran

**Affiliations:** 1Medical Oncology Department, Catalan Institute of Oncology Badalona, Hospital Universitari Germans Trias i Pujol, Badalona Applied Research Group in Oncology, Institut Germans Trias i Pujol, 08916 Barcelona, Spain; 2Division of Immunology, Laboratori Clinica Metropolitana Nord, Hospital Universitari Germans Trias i Pujol, Institut Germans Trias i Pujol, 08916 Barcelona, Spain; 3Department of Cellular Biology, Physiology and Immunology, Universitat Autònoma de Barcelona, 08916 Barcelona, Spain; 4Research Nurse Team, Catalan Institute of Oncology Badalona, Hospital Universitari Germans Trias i Pujol, 08916 Barcelona, Spain; 5Internal Medicine Department, Addiction Unit, Hospital Universitari Germans Trias i Pujol, 08916 Barcelona, Spain; 6Department of Medicine, Universitat Autònoma de Barcelona, 08916 Barcelona, Spain

**Keywords:** SARS-CoV-2, lung cancer, vaccination immune response, anti-spike antibodies

## Abstract

**Simple Summary:**

During the SARS-CoV-2 pandemic, lung cancer patients have been considered an especially vulnerable population and have been prioritized for vaccination. However, several aspects (degree of immunity, potential interaction with active anticancer therapy, safety, and tolerability of the vaccines) remained unclear. We sought to evaluate the immune response to vaccines in this population and detail vaccine-related adverse events. In our cohort of 126 lung cancer patients, SARS-CoV-2 vaccines were safe irrespective of the systemic therapy, and vaccine-related adverse events and efficacy were similar regardless the age. Most of the patients developed SARS-CoV-2 antibodies after first and second dose of the vaccine, which was maintained over time. Rates of infection after vaccination were low, more frequent with the Omicron variant, with a milder clinical course after vaccination. The rate of hospital admissions due to COVID-19 infection was very low, and no COVID-19-related deaths occurred in our cohort of patients.

**Abstract:**

Lung cancer patients represent a subgroup of special vulnerability in whom the SARS-CoV-2 infection could attain higher rates of morbidity and mortality. Therefore, those patients were recommended to receive SARS-CoV-2 vaccines once they were approved. However, little was known at that time regarding the degree of immunity developed after vaccination or vaccine-related adverse events, and more uncertainty involved the real need for a third dose. We sought to evaluate the immune response developed after vaccination, as well as the safety and efficacy of SARS-CoV-2 vaccines in a cohort of patients with lung cancer. Patients were identified through the Oncology/Hematology Outpatient Vaccination Program. Anti-Spike IgG was measured before any vaccine and at 3–6-, 6–9- and 12–15-month time points after the 2nd dose. Detailed clinical data were also collected. In total, 126 patients with lung cancer participated and received at least one dose of the SARS-CoV-2 vaccine. At 3–6 months after 2nd dose, 99.1% of baseline seronegative patients seroconverted and anti-Spike IgG titers went from a median value of 9.45 to 720 UI/mL. At the 6–9-month time point, titers raised to a median value of 924 UI/mL, and at 12–15 months, after the boost dose, they reached a median value of 3064 UI/mL. Adverse events to the vaccine were mild, and no SARS- CoV-2 infection-related deaths were recorded. In this lung cancer cohort, COVID-19 vaccines were safe and effective irrespective of the systemic anticancer therapy. Most of the patients developed anti-Spike IgG after the second dose, and these titers were maintained over time with low infection and reinfection rates with a mild clinical course.

## 1. Introduction

SARS-CoV-2 infection during a period of immunosuppression induced by chemotherapy (ChT) or immune modulation induced by other antitumor treatments can be accompanied by a significant morbidity and mortality rate in cancer patients [1,2].

Lung cancer (LC) patients also present other risk factors that make them particularly vulnerable such as age at the presentation at ≥70 years, previous respiratory disease, other comorbidities generally associated with smoking history, diagnosis in advanced stages in most cases, and the intrinsic prognosis of the disease [3,4]. With the approval of different vaccines against SARS-CoV-2, Oncology Scientific Societies recommended that cancer patients were prioritized to receive these vaccines [5,6,7,8,9]. Despite that, cancer patients did not participate in the development of these vaccine studies [10,11,12,13].

These recommendations, therefore, raised many questions in this group of patients, such as the degree and duration of immunity generated after vaccination, whether the adverse events (AE) may be greater than in the general population, and if vaccination may interfere with the efficacy of antitumor treatments and patient survival. Additional uncertainty involved the evidence for a third dose of the vaccine in this population once this was recommended.

Previous data in patients without cancer suggest an association of humoral immunity with the severity of the infection and point to other mechanisms, beyond antibodies to SARS-CoV-2, as key elements in protection against this virus [14].

Effective measures, such as small-molecule inhibitors, bioactive natural products, and traditional medicine, are greatly needed to reduce SARS-CoV-2 transmission [15,16,17,18]. However, promising magic bullets still do not exist. As an indispensable resource, vaccines have demonstrated potential value in countering SARS-CoV-2 infection [19,20].

We hypothesized that vaccination against SARS-CoV-2 may trigger a different immune response in LC patients depending on a history of the previous infection and the anticancer treatment they were receiving at the time of vaccination. Vaccine-related AE could be modified during such treatments, as well as the expected treatment-derived AE. Potentially there could be an impact on the treatment efficacy.

Our main objective was to evaluate the immune response against the SARS-CoV-2 vaccine in LC patients by pre- and post-vaccination (at 3–6, 6–9, and 12 months) determination of IgG against SARS-CoV-2 in this cohort of LC patients.

Secondary objectives included the evaluation of whether the AE rate of the vaccines is similar to that reported in the registry studies and the (re)infection rates after vaccination as well as complications and mortality related to SARS-CoV-2 infection. This objective was of special interest when the Omicron variant dominated prior variants of concern. In addition, special interest was given to the population aged ≥75 years, in whom the efficacy and AE vaccine-related information were limited.

## 2. Material and Methods

This was a prospective study for the evaluation of immune response after vaccination against SARS-CoV-2 in patients with LC. All patients, regardless of stage or active treatment, with or without known previous infection with SARS-CoV-2, were candidates for the study. Patients should be available for clinical and serological follow-up. Approval was obtained from the Institutional Review Board of Hospital Universitari Germans Trias i Pujol (COVID Lung Vaccine Study on behalf of Seroncovid Study PI-20-202). Informed consent was obtained from all the participants. 

A vaccination program was set up at our Oncology/Hematology Outpatient Clinic. The priority sequence for LC patients was as follows: (1) patients on TKI or radiotherapy; (2) patients on IT; (3) patients on ChT or ChT-IT; and (4) patients on surveillance. This priority sequence was selected considering both the risk related to disease burden and stage and the potential interactions of vaccines and systemic cancer therapies. Patients ≥80 were vaccinated at any time, irrespective of active therapy, if they had not been already vaccinated through the Primary Care Program, considering that people aged ≥80 were the first group of people vaccinated in our country.

A pre-vaccination IgG determination was performed and allowed to identify the following groups of patients: patients with known previous SARS-CoV-2 infection with IgG+ (group 1), patients with known previous SARS-CoV-2 infection with IgG− (group 2), patients with unknown previous SARS-CoV-2 infection with IgG+ (asymptomatic course, group 3), and patients with no known previous SARS-CoV-2 infection with IgG− (group 4). The sample was collected in a 10 mL serum tube. After vaccination, IgG determination was repeated at 3–6, 6–9, and 12 months. We arbitrarily proposed such time points to try to fully characterize the degree and duration of immunity in our cohort of patients. 

Information on immediate vaccine-related AE was collected through calls at 24, 48, and 72 h post-vaccination. Information regarding long-term AE was also registered (Figure 1).

Clinical information was collected from the patients’ medical records. All patients were followed up during their usual visits, and whenever possible, serological extractions were performed, taking advantage of blood draws necessary for their oncologic care. Some of the controls were performed using the virtual infrastructure established in our center (video call or telephone call) during the pandemic. Special interest was given to the population aged ≥75 years, where information on the general population is very limited. Clinical information included demographic data, smoking and use of other substances; comorbidities (especially information related to previous respiratory pathology), tumor-related information, vaccine-related information; serological information by pre- and post-vaccination IgG determination; post-vaccination SARS-CoV-2 (re)infection; and survival information.

The inclusion in this study was subjected to the start of the vaccination campaign determined by the Regional Health Authorities. At the initial stages of this study, we could not foresee the impact of both the recommendation of a third dose and the Omicron variant influence in terms of community transmission. These facts made us slightly modify the blood draws calendar to consider the third dose administration when possible. 

### 2.1. Serologies

Serum IgG anti- Spike levels were determined by a quantitative ELISA (COVID-19 quantitative IgG ELISA, Demeditec Diagnostis^®^). A 5-Parameter Logistic (5PL) curve was built with known standards, and antibody titer information was obtained from the samples using an automated ELISA analyzer. These analyses were performed in the Immunology laboratory of HUGTIP. Anti-Spike antibodies titers were considered positive if >40 UI/mL and negative if <32 UI/mL. Patients with values between 40 UI/mL and 32 UI/mL were considered equivocal and were retested. If the result was equivocal again, the sample was considered negative.

### 2.2. Statistical Considerations

Qualitative variables were calculated as percentages, means, medians, SDs, and ranges. Quantitative variables were calculated as percentages. For the main analysis, qualitative variables were compared with the Kruskal–Wallis test for independent samples. 

All the patients were followed until death, withdrawal of consent, or loss of follow-up. Patients who were still alive at the date of the last contact were censored (censoring data 2 June 2022).

Deidentified data were exported from Microsoft Excel version 2013 for Windows (Microsoft Corporation, 2013). Statistical analyses were performed with Statistical Package for the Social Sciences (SPSS Inc. Chicago, IL, USA) version 24 for Microsoft Windows. Statistical significance was determined when *p* ≤ 0.05.

## 3. Results

### 3.1. Cohort Description

From 31 March to 15 May 2021, 126 patients participated in the study. In total, 61.9% were male, with a median age of 66 y (46–83), 88.1% were Non-Small-Cell Lung Cancer (NSCLC), and 76% had stage IV at diagnosis. Systemic therapy included *EGFR/ALK/ROS1/RET/MET* oral inhibitors (19.8%), immunotherapy (IT) (41.8%), IT-ChT (14.1%), and ChT (19.9%). Nine patients were on active surveillance (Table 1). TKI included erlotinib, afatinib, osimertinib, and alectinib. IT included pembrolizumab, atezolizumab, nivolumab, and durvalumab. ChT included platin-containing regimens. A total of 48 patients were receiving steroids at the time of vaccination, and 47 patients as ChT premedication and/or antiemetic. Only one patient in the 1st line TKI group was receiving high-dose steroids due to CNS progression being treated with Whole Cranial Radiation. None of the patients were receiving other immune-suppressors or GSCF along with the chemotherapy. 

The cohort of patients ≥75 included 19 patients, with 68.4% male, with a median age of 77 (75–83). A total of 73.7% were NSCLC, and 68.4% had stage IV at diagnosis. Systemic therapy in this cohort included TKI, IT, IT-ChT, and ChT in 26.3%, 10.5%, 21.1%, and 26.3%, respectively. Two patients (10.5%) were on active surveillance (Table 1). 

### 3.2. Type of Vaccines 

Out of 126 patients, 95.2% received Moderna mRNA-1273 (Moderna^®)^) on behalf of the Oncology/Hematology Outpatient Vaccination Program as 1st and 2nd dose. Only one patient in the cohort (0.8%) passed away after 1st dose and did not receive subsequent doses. The second dose included Moderna^®^ in 119 (94.4%), Pfizer BNT1612b (Pfizer^®^) in 2 (1.6%), and Astra-Zeneca hAdOx1 (Astra-Zeneca^®^) in 4 (3.2%) patients (Table 2). According to our National vaccination recommendations, the boost should include Moderna^®^ being Pfizer^®^ reserved for those patients aged ≥75, always considering the vaccine’s availability. Thirty patients did not receive the third dose (27 (90%) due to cancer-related death, and 3 (10%) due to patient choice). A total of 96 patients received the 3rd dose, 93 (96.8%), 1 (1.1%), and 2 (2.1%) patients received Moderna^®^, Pfizer^®^, and Astra-Zeneca^®^, respectively. 

In total, 17 (89.5%) and 2 (10.5%) patients ≥75 received Moderna^®^ and Pfizer^®^ as 1st and 2nd dose, respectively. The third dose included Moderna^®^ and Pfizer^®^ in 13 (92.8%) and 1 (7.1 %) patient, respectively. 

### 3.3. Prior Serologic Status 

Evaluable baseline blood samples were available from 122 patients. Thirteen patients had SARS-CoV-2 infection prior to vaccination. At baseline, 7 patients (5.7%) had a symptomatic infection, with positive baseline IgG in 4 of them (57.1%, group 1). Three patients did not develop IgG after confirmed infection (42.8%, group 2). No vaccine-related AE was reported in any of these groups. Four additional patients had positive baseline IgG without prior symptomatic infection (group 3). The rest of the patients (no documented infection and negative baseline IgG) were included in group 4 (*n* = 106) (Figure 1).

### 3.4. Seroconversion 

One hundred and nine patients with negative baseline IgG (groups 2 and 4) were considered for the calculation of seroconversion rates after 1st and 2nd dose. All patients except one seroconverted (99.1%) at 3–6 time point. For the entire cohort, baseline median IgG titers were 9.45 UI/mL (0.05–690) in 122 patients and increased to 720 UI/mL (9.91–8169) at 3–6-month time point in 81 patients, 924 UI/mL (80–1153) at 6–9-month time point in 65 patients and 3064 UI/mL (674.7–4112.6) at 12–15-month time point in 83 patients. Differences between each time point were significant (*p* < 0.01) (Figure 2A).

Three out of four patients in group 1 patients experienced an increase in IgG levels after 1st and 2nd vaccines (no data in patient #4, who had median IgG levels of 801 UI/mL (477–3399) vs. 481.5 UI/mL (50.7–690) at baseline).

Two out of three patients in group 2 seroconverted with median titers of 1367.5 UI/mL. One out of these two patients maintained high titers of IgG throughout the study (1038 and 3309.5 at 6–9- and 12–15-month time points, respectively). The other two patients did not have additional IgG data since they passed away due to cancer progression.

Five out of nine patients in group 3 patients experienced an increase in IgG levels after 1st and 2nd vaccine (4 patients had no sample, 2 due to death and 2 due to no sample available) at 3–6-month time point with a median IgG titers of 1500 UI/mL (390, 8169). Median baseline IgG titers were 384 UI/mL (49.4–690). Four patients maintained those IgG levels at 6–9-month time point (5 patients had no sample, 3 due to death and 2 due to no sample available) with median titers of 1084 UI/mL (642, 2373). Six patients maintained high titers of IgG at the 12–15 time point [2849.9 UI/mL (1950.9–3391.8)]. Three patients in this group had passed away due to cancer progression.

Median IgG baseline titers for group 4 patients (*n* = 106) were 9.04 UI/mL (0.05–32.5). All patients except one seroconverted in this group (99.14%) with median IgG titers of 692 UI/mL (56–3110) at 3–6 time point. Titers of IgG were sustained with median values of 882 UI/mL (9.91–8169) and 3035 UI/mL (80–3969) at 6–9 and 12–15 months, respectively. 

Twelve-to-fifteen-month time point include serological data from 81 patients (36 passed away, and 5 were not available for blood draws). Three patients did not receive the 3rd dose, and the median time since 2nd dose was 311.3 days (308–317) with median titers of 3507.5UI/mL (3308.1–3802.8). Two of these patients had had pauci-symptomatic SARS-CoV-2 infection. The median time since 3rd dose to 12–15 time point for the rest of the patients (*n* = 79) was 118.38 days (41–404) with median IgG titers of 2793.6 UI/mL (674.7–4112.65)

When considering patients ≥75 (*n* = 19), 9 out of 15 seroconverted after 1st and 2nd doses (4 patients were excluded due to positive baseline IgG, and 3 patients did not have a sample at 3–6 months since they passed away). Median baseline titers were 10.5 UI/mL (0.05–32.5). Titers after vaccination increased to a median of 891.2 UI/mL (111.9–1464), 1673.2 (80–3696), and 2568.6 (674.7–4019.3) at 3–6-, 6–9- and 12–15-month time point. Patients in groups 1 and 3 increased IgG titers to 4834.5 UI/mL (1500–8169) and 3019 UI/mL (2801.1–3237.1) at 3–6- and 12–15-month time point with respect to baseline [475.3 UI/mL (199–690)]. Data on 6–9-month time point were limited to one patient with titers of 1206 UI/mL. Differences in IgG titers between each time point were significant, with a similar titers trend as in the general population (*p* < 0.01) (Figure 2B).

Anticancer treatments are summarized in Table 1. By grouping the schedules according to treatment types, 23.01% were receiving ChT-containing regimens; 14.3% were receiving ChT-IT; 34.9% were receiving IT, 19.8% were receiving TKI, and 7.9% were on active surveillance (*n* = 10) or other treatments (*n* = 1, lantreotide). 

Median IgG baseline titers were 10.7 UI/mL (0.05-690) in patients receiving ChT (*n* = 29) and increased to 831 (262–1788), 865.5 (315–1788), and 3473.7 (789.3–4112.6) UI/mL at 3–6, 6–9, and 12–15 months. Median IgG baseline titers were 10.2 UI/mL (0.05–690) in patients receiving ChTIT (*n* = 17) and increased to 448.5 (123–8169), 507 (112–2373), and 2801.1 (817–3930.6) UI/mL at 3–6, 6–9, and 12–15 months. Median IgG baseline titers were 9.05 UI/mL (0.05–690) in patients receiving IT (*n* = 42) and increased to 835 (53–3399), 1221 (188–3456), and 3235.7 (983.4–3857.8) UI/mL at 3–6, 6–9, and 12–15 months. Median IgG baseline titers were 6.03 UI/mL (0.05–525) in patients receiving TKI (*n* = 24) and increased to 598 (111.9–2340), 960 (80–3696), and 2978.8 (674.7–3936.3) UI/mL at 3–6, 6–9, and 12–15 months. Median IgG baseline titers were 9.49 UI/mL (0.05-11.1) in patients on surveillance (*n* = 9) and lantreotide (*n* = 1) and increased to 639 (9.91–3267), 765 (212–1587), and 2620.5 (1540.2–3571.6) UI/mL at 3–6, 6–9, and 12–15 months. Median IgG titers significantly increased at each time point when compared with the corresponding baseline and prior time points, and this increment occurred irrespective of the therapeutic schedule or active surveillance (*p* < 0.01) (Figure 3A–E).

### 3.5. Adverse Events to Vaccines 

AES after 1st and 2nd doses were generally mild and included local pain at the vaccination site (34.9 and 35%), asthenia (6 and 8.7%), and myalgia (3.9 and 18.3%). These were slightly more frequent in patients ≥75, especially after 2nd dose (26.3 and 42.1%; 5.3 and 15.815%, and 5.3 and 42.1% for pain, asthenia, and myalgia, after 1st and 2nd dose, respectively). Most frequent AE after 3rd dose locally included pain (20.6%), asthenia (6.2%), and myalgia (7.2%). Pain, the most frequent AE after 3rd dose, occurred in 28.6% of patients ≥75 (Table 2).

Patients treated with IT and TKI had a higher percentage of AEs. Overall, AEs were recorded in 64%, 100%, and 52% of the patients on TKI with the 1st, 2nd, and 3rd dose, respectively. AEs on patients receiving any IT-containing regimen were recorded at 57.3%, 85.2%, and 39,3% with the 1st, 2nd, and 3rd dose, respectively. AEs in patients on exclusive ChT-containing schedules and on surveillance occurred in 25.7%, 34.3%, and 22.8% of the cases with the 1st, 2nd, and 3rd dose, respectively. All the AE reported in our cohort were grade 1, except for one patient on 1st line pembrolizumab who reported grade 2 asthenia after the 1st dose. The pain was the most common AEs reported by patients on TKI (44%, 48%, and 24% with the 1st, 2nd, and 3rd dose, respectively) compared to patients on any IT- containing regimen and patients on ChT and on active surveillance (36.1%, 27.8%, and 16.4% and 14.3%, 25.7% and 11.4% with 1st, 2nd and 3rd dose, respectively).

### 3.6. Protection

Infection after 2nd dose occurred in 5 patients after a median of 223 days (162–253) of the vaccine administration (Table 2). One patient had a nosocomial but asymptomatic infection, three patients were pauci-symtomatic, and one patient was admitted to the hospital due to bilateral pneumonia. He received steroids, antiviral therapy with remdesivir, and oxygen. This patient had a stage IV squamous cell carcinoma, a good performance status (PS1), and was receiving 2nd line with IT. However, past history included COPD, diabetes, hypertension, and chronic renal failure. The patient was discharged after 10 days of admission and fully recovered from the infection. This patient seroconverted at 3–6-month time point. 

Infection after 3rd dose occurred in ten patients after a median of 77 days (26–149) of the vaccine administration. Eight patients were infected in the context of family transmission, and 2 patients tested positive while admitted to the hospital but did not develop any SARS-CoV-2 infection symptoms. Interestingly, one patient belonged to group 3. She was infected during the 1st waves of the pandemic and had a baseline IgG titer of 57 UI/mL, which increased at 3–6-month time point and did not decline over time (titers were of 583 UI/mL at 3–6, 963 UI/mL at 6–9 (after symptomatic infection) and 1950.9 UI/mL at 12–15-month time points (before nosocomial infection). 

Overall, reinfection occurred in 2 patients (1.6%) < 75 years old and coursed with mild symptoms. 

### 3.7. Mortality 

The mortality rate in our cohort was 30.2% (27.8% due to cancer progression and 2.4% due to other causes, including hemorrhagic stroke in one case and other causes in two patients). No SARS-CoV-2-infection-related deaths were recorded in this cohort of LC patients (Table 2). 

## 4. Discussion

Patients with LC represent a special vulnerable population in terms of risk of poor outcomes if infected by SARS-CoV-2. The implementation of the vaccination campaigns has been a turning point in the control of infection and its severity, in addition to the effect of social distancing and other measures. If cancer patients, especially those on active systemic therapy, may experience a reduced efficacy of the administered vaccine was a matter of concern once the Vaccination Campaigns started at the beginning of 2021 in our country. Mirroring prior evidence of lower antibody responses and protection to influenza vaccines in individuals with cancer, different initiatives sought to understand the immune response to SARS-CoV-2 vaccines in those patients [21,22,23,24,25].

SARS-CoV-2-infected patients produce IgG that binds viral S and N proteins. Antibodies that recognize the RBD of S protein are particularly relevant for their neutralizing viral capacity. These antibodies bind to RBD and impair the interaction with the ACE2 protein, the receptor molecule on target cells, thus preventing the virus entry [26]. However, the impact of qualitative differences and their changes in disease severity had not been fully elucidated. As it occurred after SARS-CoV-2 infection, in addition to the induction of cellular immunity, SARS-CoV-2 vaccines stimulate the formation of virus-neutralizing antibodies [27]. Prior studies have demonstrated an impaired neutralizing IgG response to the mRNA vaccine in cancer patients. Most of them were over-represented by patients with hematologic malignancies, being patients with solid tumors a minority [25,28].

Despite the fact that current anti- SARS-CoV-2 vaccines are not nasal vaccines, the presence of IgG and IgA neutralizing antibodies has been demonstrated in the mucosa of patients receiving current intra-muscular vaccines. This suggests an in situ production of these antibodies or a transcytosis migration from circulating immunoglobulins [29], but antibody protection goes far beyond its neutralizing capacity. The ability of antibodies to activate the complement, which is a basic innate mechanism in pathogen defense, has well extensively described. This pathway has been demonstrated to be protective against SARS-CoV-2 [30]. In addition, once attached to their targets, antibodies opsonize the pathogens in order to be phagocyted by macrophages and monocytes [31]. This indicates that, although most of the studies have focused on neutralizing antibodies, all IgG species against SARS-CoV-2 exert different protective functions against the virus.

Then, when evaluating this subgroup of patients with solid tumors (LC represents 14–18%), no significant associations with reduced neutralizing antibodies were found (including cancer subtype and stage, type of anticancer therapy) beyond the lack of the previous infection and older age [25]. In addition, some studies have evaluated the IgG response shortly after the second dose with limited data on the third dose effect. In such studies, IgG titers appeared to be lower in patients receiving ChT, IT, and TKI compared to those under surveillance or healthy controls. In such studies, LC patients represent a 20–25% serologic response after the vaccine appeared to indicate a relatively conserved antibody production with no impact when patients received antiPD(L)1-based IT [28]. Some studies have found that chronic corticosteroid treatment, as well as age and chemotherapy as the last systemic treatment within 3 months prior to vaccination, were associated with a lack of immunization [24]. Since the use of high-dose steroids was low in our cohort, no effect was observed in this regard. 

Long-lasting longitudinal follow-up in our study shows no impact on IgG titers according to anticancer therapy. All subgroups of patients appeared to seroconvert with an adequate median of IgG titers. Most of the patients in our cohort mounted adequate IgG titers in response to vaccination after two doses. Only one patient was infected between the first and second dose. IgG titers were maintained over time in evaluable patients. For those patients receiving the 3rd dose, the rise of IgG titers suggests that repeated vaccination is crucial in maintaining IgG titers in patients with LC and would support future boosts of the vaccine. However, the adequate timeline for additional boosts remains unclear.

Several SARS-CoV-2 vaccines were approved and recommended by the US Food and Drug Administration and the European Medicines Agency following the results of phase 3 trials. Most showed a 95–94% efficacy, with lower efficacy for other vaccines [10,11,12,13]. In addition, they demonstrated a good safety profile in terms of both local and systemic AEs. The most common local reaction was pain, and the most common systemic AEs were fatigue, headache, and myalgias. Most of them were grade 1 or 2 and increased after the 2nd dose. AEs were more common among younger participants.

Some studies have evaluated the safety of vaccines in patients with solid tumors, but LC represents 13% to 25% of the total. AEs were similar in cancer patients to those reported from trials in healthy individuals [30,31,32]. In our cohort, we found similar types and grades of AEs, with slightly higher in older patients. Of note, the definition of older patients for registry trials ranged from ≥55–65 years, while we established ≥75 as the cut-off point. Other studies focused on LC did not report any differences in AE according to age [24]. 

The relationship between AEs and the specific type of anticancer treatment has been unexplored in patients with cancer. Patients on IT and TKI in our cohort experienced a higher percentage of AEs with no grade differences. 

Beyond common and short-term AE to SARS-CoV-2 vaccination, efforts have been made to evaluate the medium- and long-term risks of the vaccine. Vaccination has been strongly associated to the risk of several AE such as myocarditis, lymphadenopathy, appendicitis, and herpes zoster infection with BNT162b2 vaccine over a 42-day follow-up period after vaccination, Bell’s palsy, and thromboembolic events associated to adenoviral vector vaccines (ChAdOx1 nCoV-1932 and Ad26.COV2.S) [32,33,34]. These AE occurred in a period of time relatively short from the vaccine shot. The follow-up period of our study has been enough to rule out the occurrence of short and medium-term AE.

However, additional studies to estimate the potential risk of medium- and long-term toxicity are needed. Moreover, these short and medium-term AE need to be considered in the context of additional boosts. 

In the general population, the reinfection rate of SARS-CoV-2 is relatively low (0.65%), with an even lower symptomatic reinfection rate (0.37%), being the protection against SARS-CoV-2 after natural infection comparable to that estimated for vaccine protection [33]. Specific data on reinfection rates in cancer patients are scarce. Prior series have communicated reinfection rates after vaccination of 2.6% [24]. Recent studies have demonstrated an association between antibody titers and vaccine efficacy. However, a correlate of protection (CoP), an immunological marker associated with protection against infection, for SARS-CoV-2 remains currently undefined. A recent review suggests that a SARS-CoV-2 CoP is a likely relative, with higher antibody levels decreasing the risk of infection but without a complete elimination [35]. Moreover, the emergence of variants of concern (VOCs) further complicates the search for a SARS-CoV-2 CoP. Data from several studies raise the possibility that a SARS-CoV-2 CoP may be VOC-specific and reported that vaccinated subjects have reduced neutralizing ability against VOCs, including Beta (B.1.351), Delta (B.1.617.2) and Omicron (B.1.1.529) [36,37]. Our study started in March 2021, when Alpha (B1.1.7) was the main variant in Spain, and finished in June 2022 after the appearance of Delta and Omicron variants of concern. In recent studies, correlate of protection for WT and alpha strain were established at 154 and 168 BAU/mL, respectively, far lower than the median of antibodies observed in our cohort. As successive variants, such as Delta and Omicron, incorporated new mutations able to escape antibody response, this parameter must be re-assessed. Recent data suggest that booster doses are able to induce neutralizing antibodies against these new variants [38,39,40,41].

In our cohort, rates of infection and reinfection after vaccination were low, more frequent with Delta and Omicron variants, and with milder clinical course after vaccination. The rate of hospital admissions due to SARS-CoV-2 infection was very low, and no COVID-19-related deaths were annotated. Such results confirm the efficacy of the SARS-CoV-2 vaccines in LC patients in whom the risk of mortality due to SARS-CoV-2 infection was reported at about 30% in the initial reports [4,42,43]. Thus, vaccination prevents both the severity of the infection and the risk of death in this vulnerable population.

To the best of our knowledge, this is one of the largest cohorts of patients with LC, which includes not only a long-lasting longitudinal follow-up with a majority of patients having received the full vaccination plus a booster vaccine dose but also information related to the infection and reinfection rates due to novel variants of concern, such as Delta and Omicron, which are underrepresented in other studies [25,43]. Moreover, this is a cohort that only included patients with LC, most of them under active treatment due to advanced or locally advanced stages, which represent the main clinical scenarios an Oncologist may face in their daily practice. Conversely to other studies, patients on surveillance included in our study represent a minority [24]. 

Limitations of our study included, first, the dropout rate throughout the study, which did not allow a full serologic analysis. However, it was consistent with the cancer-related general condition and deaths that can occur in a cohort of patients with advanced LC. Second, anti-spike Ig titers have shown a good correlation with virus neutralization; however, some studies have demonstrated variable neutralizing activity despite high anti-spike Ig titers depending on the SARS-CoV-2 strain [44,45]. Cellular COVID-19 studies must be performed in fresh blood samples. They can also be performed in frozen cellular samples but with lower sensitivity. In our study, we only obtained and stored serum samples. Therefore, unfortunately, cellular studies cannot be performed. However, the presence of antibody response is indicative of a cellular T lymphocyte response, as T-cell response is necessary for IgG anti-protein B cell generation. The lack of detection of T-cell response in a seropositive patient is probably due to sensitivity issues of the technique [46]. Despite the fact that our study did not include neutralizing activity evaluation, the low infection rates, as well as the severity of the infection, suggest good protection against the infection in this cohort of patients. Third, one concerning question is how to distinguish the antibody production in those vaccinated patients who were prior naturally infected by the SARS-CoV-2. As seroconversion occurs between 7 and 14 days after infection, the infected patients that did not exhibit positive anti-Spike antibodies at baseline were supposed not to have developed natural antibodies against the virus [47]. The Anti-Spike IgG levels obtained by natural infection are, on average, one logarithm lower than those obtained by vaccination. The Anti-Spike IgG levels observed in our cohort, either in the seropositive or the seronegative baseline infected patients, are in the range of vaccinated patients and can be mainly attributed to the vaccine response [48]. Fourth, having obtained an extra sample before the vaccine boost could have helped us to identify some patients with potentially declining IgG titers and more likely to benefit from this boost. However, on behalf of the National Vaccination Program, no changes in recommendations would have been derived from such information. Fifth, the classification into four groups may fragment the sample since, as expected, groups 1–3 represent small numbers. However, such subdivision has the only purpose of fully characterizing our cohort of patients in terms of pre-vaccination status, to try to understand if the prior serologic status could modify the ability to seroconvert. Given the own nature of our study, the addition of new patients to the smaller subgroups is not possible since, as stated, the current rate of vaccination in our environment is very high. Last, the lack of an unvaccinated control group to compare infection rates and the severity of the infection to firmly conclude adequate protection. Since cancer patients were considered a high-risk group, they were prioritized to receive the SARS-CoV-2 vaccine. In addition, in our particular population, adherence to vaccine recommendations was extremely high. This fact made it almost impossible to achieve a similar but non-vaccinated population group to compare with. Our cohort included a population of 126 patients. However, after retrospectively reviewing clinical records of a total of 432 patients with lung cancer, only 9 patients refused the vaccine. One out of nine patients passed away due to SARS-CoV-2-related pneumonia, two patients were infected by Omicron with a mild symptomatic course, and the rest have had no symptoms of SARS-CoV-2 infection. We consider that those numbers are too small to draw any meaningful conclusion. In addition, the lack of a control unvaccinated group precludes verifying if the vaccination has altered the mortality patterns in this cohort of LC patients.

## 5. Conclusions

In conclusion, in our cohort of LC patients, SARS-CoV-2 vaccines were safe irrespective of the systemic anticancer therapy, and AES and efficacy were similar regardless the age. Most of the patients developed immunity after the first and second dose. IgG titers were maintained over time with low infection and reinfection rates with a mild clinical course. With the emergence of novel variants of concern, tailoring vaccine regimes for cancer patients might be a matter of interest. 

## Figures and Tables

**Figure 1 cancers-15-00137-f001:**
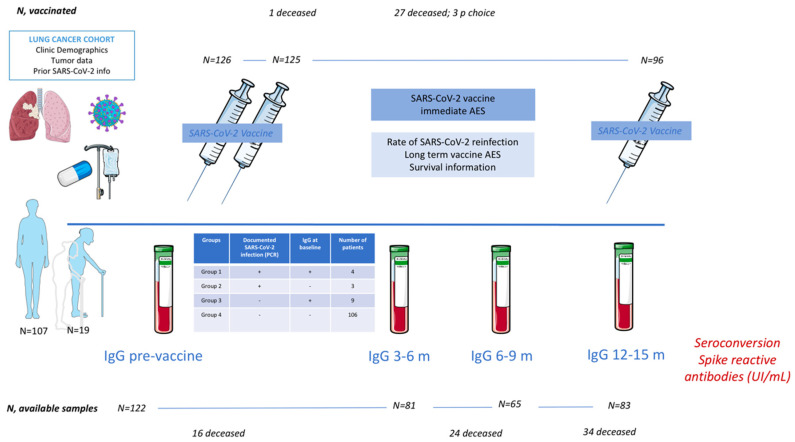
Effect of COVID-19 vaccination in lung cancer patients: COVID Lung Vaccine Study Scheme and patient disposition. Legend: AEs: adverse events; IgG, Immunoglobulin G; n, number of patients; PCR, polymerase chain reaction.

**Figure 2 cancers-15-00137-f002:**
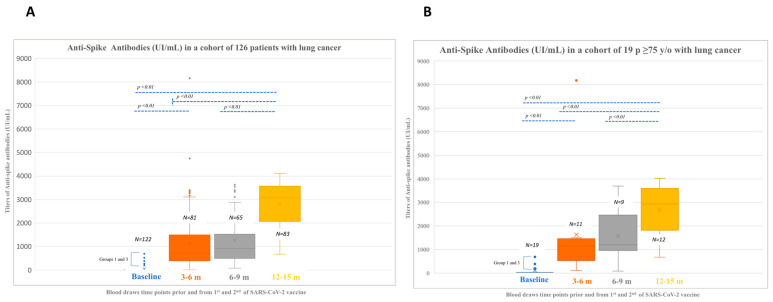
(**A**). Anti-spike antibodies titers in a cohort of 126 patients with lung cancer. (**B**). Anti-spike antibodies in the subgroup of 19 patients with lung cancer aged ≥75. (baseline samples were evaluable from 122 and 19 patients from groups A and B, respectively). Legend: m, months.

**Figure 3 cancers-15-00137-f003:**
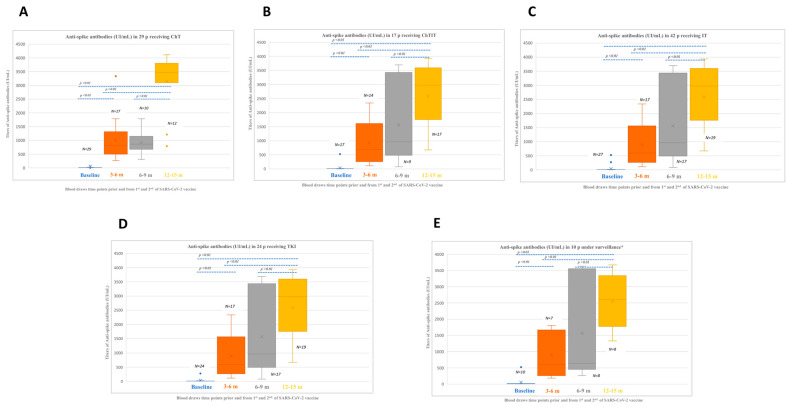
Anti-spike antibodies titers disposition according to type of anticancer therapy in a cohort of 126 patients with lung cancer (baseline samples were evaluable from 122 patients). (**A**). Anti-spike antibodies in 29 patients receiving ChT; (**B**). Anti-spike antibodies in 17 patients receiving ChTIT; (**C**). Anti-spike antibodies in 42 patients receiving IT; (**D**). Anti-spike antibodies in 24 patients receiving TKI; (**E**). Anti-spike antibodies titers in 10 patients on surveillance *; Legend: ChT, chemotherapy; IT, immunotherapy; m, months; p, patients; RT, radiotherapy; TKI, tyrosine kinase inhibitor. * This group includes 9 patients on active surveillance and 1 patient receiving lantreotide.

**Table 1 cancers-15-00137-t001:** Demographic and tumor-related variables of a total of 126 lung cancer patients and 19 patients aged ≥75 included in the study. Legend: 1L first line: 2L second line, ChT, chemotherapy; IT, immunotherapy; LCNEC, large cell neuroendocrine carcinoma; NSCLC, non-small cell lung cancer; PS, performance status; RT, radiotherapy; SCLC, small cell lung cancer; TKI, tyrosine kinase inhibitor. * Other includes lantreotide. Bold: specific information on population >75 starts.

Overall Population	N = 126
Age [median, (range)]	66 (46–83)
Variable	N (%)
Sex Male Female	78 (61.9%) 48 (38.1%)
PS PS0 PS1 PS2	39 (30.9%) 75 (59.5%) 12 (9.5%)
Histology NSCLC LCNEC Carcinoid SCLC	111 (88.1%) 2 (1.6%) 1 (0.8%) 12 (9.5%)
Stage I–II IIIA IIIB IV	6 (4.8%) 4 (3.2%) 20 (15.9%) 96 (76.2%)
Molecular alterations *EGFR* *ALK* *BRAF* *ROS1* *MET* *RET*	16 (12.7%) 5 (3.9%) 2 (1.6%) 1 (0.8%) 1 (0.8%) 1 (0.8%)
Treatment 1L ChT 2L ChT Adjuvant ChT 1L ChTIT Adjuvant ChTIT Neoadjuvant ChTIT 1L IT 2L IT Consolidation IT Other IT lines ChTRT 1L TKI 2L ITK Other * Active surveillance	17 (13.5%) 5 (3.9%) 3 (2.4%) 16 (12.7%) 1 (0.8%) 1 (0.8%) 17 (13.5%) 20 (15.9%) 6 (4.8%) 1 (0.8%) 4 (3.2%) 20 (15.9%) 5 (3.9%) 1 (0.8%) 9 (7.2%)
**Population ≥75 year old**	**N = 19**
Age [median, (range)]	77 (75–83)
Variable	N (%)
Sex Male Female	13 (68.4%) 6 (31.6%)
PS PS0 PS1 PS2	4 (21.1%) 13 (68.4%) 2 (10.5%)
Histology NSCLC Carcinoid SCLC	14 (73.7%) 1 (5.3%) 4 (21.1%)
Stage IIIA IIIB IV	2 (10.5%) 4 (20.1%) 13 (68.4%)
Molecular alterations *EGFR* *ALK* *KRAS*	4 (21.1%) 1 (5.3%) 1 (5.3%)
Treatment 1L ChT 2L ChT 1L ChTIT 1L IT Consolidation IT 1L TKI 2L TKI Other * Active surveillance	3 (15.8%) 2 (10.5%) 4 (21.1%) 1 (5.3%) 1 (5.3%) 4 (21.1%) 1 (5.3%) 1 (5.3%) 2 (10.5%)

**Table 2 cancers-15-00137-t002:** Data regarding SARS-CoV-2 infection and vaccine-related variables of a total of 126 lung cancer patients and 19 patients ≥ 75 years old included in the study. Bold: specific information on population >75 starts.

Overall Population	N = 126
Variable	N (%)
Prior infection Group 1 (infection + IgG+) Group 2 (infection + IgG−) Group 3 (UK infection IgG+) Group 4 (UK infection IgG−)	4 (3.2%) 3 (2.3%) 9 (7.1%) 110 (87.3%)
Type of SARS-CoV-2 vaccine (1st/2nd doses) Moderna^®^ Others	120 (95.2%) 6 (4.8%)
Local adverse events after 1st dose Local pain at the vaccine administration site (grade 1) Local inflammation at the administration site (grade 1/grade 2)	44 (34.9%) 2 (1.6%)/1(0.8%)
General adverse events after 1st dose Fever (grade 1) Muscle pain (grade 1) Asthenia (grade 1/grade 2) Headache (grade 1)	8 (6.4%) 5 (3.9%) 6 (4.8%)/1 (0.8%) 1 (0.8%)
Local adverse events after 2nd dose Local pain at the vaccine administration site (grade 1) Local inflammation at the administration site (grade 1)	46 (35%) 2 (1.6%)
General adverse events after 2nd dose Fever (grade 1) Muscle pain (grade 1) Articular pain (grade 1) Arthomialgias (grade 1) Asthenia (grade1) Headache (grade 1) Regional lymph node enlargement General rash (grade 2)	21(16.6%) 13 (18.3%) 1 (0.8%) 2 (1.6%) 11 (8.7%) 3 (2.4%) 1 (0.8%) 1 (0.8%)
Type of SARS-CoV-2 vaccine (3rd dose was administered to 96 patients) Moderna^®^ Others	93 (96.8%) 3 (3.2%)
Local adverse events after third dose Local pain at the vaccine administration site (grade 1) Local inflammation at the administration site (grade 1)	16 (16.6%) 1 (1.05%)
General adverse events after third dose Fever (grade 1) Muscle pain (grade 1) Asthenia (grade1) Headache (grade 1) Diarrhea (grade 1)	9 (9.4%) 6 (6.25%) 5 (5.2%) 2 (2.1%) 1 (1.05%)
Infections after vaccination Rate of infection Patients infected after 1st dose Patient infected after 2nd dose Patients infected after 3rd dose Rate of reinfection Patient with prior COVID-19 infection who were re-infected after vaccination Group 1 (infection + IgG+) Group 2 (infection + IgG−) Group 3 (UK infection IgG+) Severity of infection Asymptomatic (nosocomial) Mild symptoms Severe symptoms	1 (0.8%) 5 (3.9%) in 4 p 2nd dose was the last dose 10 (7.9%) 2 p reinfection 0 1 (0.8%) 1 (0.8%) 3 12 1
Survival outcome Cancer-progression-related deaths COVID-19-related deaths Other disease or COVID-19-unrelated deaths Alive	35 (27.8%) 0 3 (2.4%) 88 (69.8%)
**Population ≥75 years old**	**N = 19**
Variable	N (%)
Documented prior infection Yes No	1 (5.3%) 18 (94.7%)
Type of SARS-CoV-2 vaccine (1st/2nd doses) Moderna^®^ Others	17 (89.5%) 2 (10.5%)
Local adverse events after first dose Local pain at the vaccine administration site (grade 1)	5 (26.3%)
General adverse events after first dose Fever (grade 1) Muscle pain (grade 1) Asthenia (grade1)	1 (5.3%) 1 (5.3%) 1 (5.3%)
Local adverse events after second dose Local pain at the vaccine administration site (grade 1) Local inflammation at the administration site (grade 1)	8 (42.1%) 1 (5.3%)
General adverse events after second dose Fever (grade 1) Artromialgias (grade 1) Asthenia (grade1)	2 (10.5%) 8 (42.1%) 3 (15.8%)
Type of SARS-CoV-2 vaccine (3rd dose was administered to 14 patients) Moderna^®^ Others	13 (92.8%) 1 (7.1%)
Local adverse events after third dose Local pain at the vaccine administration site (grade 1) Local inflammation at the administration site (grade 1)	4 (28.6%) 1 (7.1%)
General adverse events after third dose Fever (grade 1) Muscle pain (grade 1) Asthenia (grade 1) Headache (grade 1)	1 (7.1%) 1 (7.1%) 1 (7.1%) 1 (7.1%)
Infections after vaccination Rate of infection Patients infected after 1st dose Patient infected after 2nd dose Patients infected after 3rd dose Rate of reinfection Patient with prior COVID-19 infection who were re-infected after vaccination Group 1 (infection + IgG+) Group 2 (infection + IgG−) Group 3 (UK infection IgG+) Severity of infection Asymptomatic (nosocomial) Mild symptoms	0 0 2 (14.3%) 0 1 1
Survival outcome Cancer0progression-related deaths COVID-19-related deaths Other disease or COVID-19-unrelated deaths Alive	5 (26.3%) 0 2 (10.5%) 12 (63.1%)

## Data Availability

The data presented in this study are available on request from the corresponding author.

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
