# Peer review of "Immune Response and Effects of COVID-19 Vaccination in Patients with Lung Cancer—COVID Lung Vaccine Study"

_cancers, 2022, doi:10.3390/cancers15010137_

Round 1

Reviewer 1 Report

The manuscript is promising as regards COVID-19 vaccination in LC patients in my opinion, but issues raised need to be satisfactory addressed before the paper could reach a publishable value. The issues are as listed below

  1. Title: Consider replacing the current tile with the suggested title below

 Immune Responses and Effects of COVID-19 Vaccination in Patients with Lung Cancer”

  1. Lines 26, 37, 45 and all through the manuscript: Do the authors mean “vaccine-related advert effects”?
  2. Line 27: …systematic cancer therapy…?
  3. Line 33: Insert “patients” after “cancer” and replace “represents” with “represent” so that the sentence reads “Lung cancer patients represent …..”
  4. Line 43-45 and all through the manuscript: Are the authors referring to the mean anti-spike IgG  titers values?
  5. Line 46: …SARS-CoV-2…?
  6. Line 48: “…patients developed immunity…” What do the authors mean by immunity here? Please provide measured values to allow for reproducibility of the work.
  7. Line 55: rates?
  8. Line 59-61: Please, recast the statement for clarity. E.g “…Oncology Scientific Societies recommended that cancer patients be prioritized…..”
  9. Line 83: Replace “ousted” with “dominated”
  10. Line 84 and 79: In lines 84, the authors stated that patients 75 years were given prioritized but in line 97, they are now talking of 80 years.  Please, be explicit and consistent.
  11. Lines 95-97: Were there categories of LC patients selected purposively? If yes, give reason; If no why place priority of the patient categories?
  12. Line 105: Any special reason for the timing of the IgG determination  i.e. 3-6, 6-9 and 12 months?
  13. Figure 1: “SARS-CoV-2” not “SARS-Cov-2”; Recast the legend for clarity and to reflect the content of the figure. E.g: Effect of COVID-19 vaccination in lung cancer patients: Scheme and patients’ disposition
  14. Lines 133-134: What happens if the antibody level is between 33 to 40 UI/mL?
  15. Line 150: Undefined acronym! What is NSCLC?
  16. Table 2: I suggest that the COVID-19 vaccine types used be categorised into two only: Moderna® and others. The number of patients that received Pfizer and Pfizer® and Astra-Zeneca® .
  17. Table 2: Survival outcome: What factual evidences do the authors have to prove that the deaths have nothing to do with the vaccination?
  18. 3.3 PRIOR SEROLOGIC STATUS: Why did the authors attribute all the immune responses (anti body production) to the vaccination knowing that some participants have produced antibodies as a result of natural infection to SARS-CoV-2 prior to the vaccination? Again, is it possible for patients who were naturally infected with SARS-CoV-2 but had no antibody response at the time the test was performed to develop the antibody due the infection later? If yes, how did the authors differentiated this later antibodies production due to the natural infection from that due to the vaccination?
  19. Conclusion: With the high number of deaths recorded, what scientific facts do the authors have to prove that the mortalities are not associated with vaccination and to support their conclusion?
  20. The citation and referencing styles in this manuscript are at variance with that of the journal

Author Response

We appreciate the time and effort of the reviewer. See attached our detailed response  to your questions.

Reviewer 2 Report

1. Paper presented presents a interesting and more important topic of study.

2. Organization of the paper is clear and well drafted.

3. Basic introduction, related study and results & discussions are accepted.

4. Results validate the proposed work and has strong influence on the topic of investigation.

5. Good references were cited in the paper.

Author Response

We appreciate the time and effort of the reviewer and the nice comments on our manuscript.

  1. Paper presented presents a interesting and more important topic of study.
  2. Organization of the paper is clear and well drafted.
  3. Basic introduction, related study and results & discussions are accepted.
  4. Results validate the proposed work and has strong influence on the topic of investigation.
  5. Good references were cited in the paper.

No changes are needed.

Reviewer 3 Report

As a public health emergency of international concern, the highly contagious COVID-19 pandemic has been identified as a severe threat to the lives of billions of individuals. Lung cancer, a malignant tumor with the highest mortality rate, has brought significant challenges to both human health and economic development. The article by Hernandez at al. is devoted to an important topic. The article is of good quality and clear. I recommend this paper to be published in the journal. Here are some minor suggestions:

1: To be complete, the authors should enrich the related illustration about the article on SARS-CoV-2 and lung cancer in “Introduction”.

2: It is suggested to add some background in introduction and highlight the novelty of this work clearly. For example, “Effective measures, such as small-molecule inhibitors (Nature. 2020, 586, 113; Front. Immunol. 2022, 13, 1015355), bioactive natural products (Pharmaceuticals. 2022, 15, 620; Biomedicines. 2021, 9, 689), and traditional medicine (Cell Biosci. 2021, 11, 100) are greatly needed to reduce SARS-CoV-2 transmission. However, promising magic bullets still do not exist (J Med Virol. 2022, 94, 1766-1767; Lancet Resp. Med. 2022, 10, e68). As an indispensable resource, vaccines have demonstrated potential value in countering SARS-CoV-2 infection.” This is critical to address in this manuscript, the authors should enrich this part in the revised version.

3: Take “SARS-CoV-2” for example, on Page 1, line 2, “SARS-COV-2”; Page 1, line 28, “SARS-COV2”; Page 2, line 71, “SARS-Cov2”; Table 2, “Sars-Cov2”; Page 12, line 320, “SARS-Cov-2”; Page 12, line 327, “SARS-COv-2”; Page 12, line 338, “antiSARS-COV-2”. The authors should make careful corrections.

Author Response

We appreciate the time and effort of the reviewer. Please find attached our detailed responses to your questions 
